# LMSA: Low-relation Mutil-head Self-Attention Mechanism in Visual Transformer

## Abstract

The Transformer backbone network with the self-attention mechanism as the core has achieved great success in the field of natural language processing and computer vision. However, through the self-attention mechanism brings high performance, it also brings higher computational complexity compared to the classic visual feature extraction methods. To further reduce the complexity of self-attention mechanism and explore its lighter version in computer vision, in this paper, we design a novel lightweighted self-attention mechanism: Low-relation Mutil-head Self-Attention (LMSA), which is superior than the recent self-attention. Specifically, the proposed self-attention mechanism breaks the barrier of the dimensional consistency of the traditional self-attention mechanism, resulting in lower computational complexity and occupies less storage space. In addition, employing the new mechanism can release part of the computing consumption of the Transformer network and make the best use of it. Experimental results show that the dimensional consistency inside the traditional self-attention mechanism is unnecessary. In particular, using Swin as the backbone model for training, the accuracy in CIFAR-10 image classification task is improved by 0.43%, in the meanwhile, the consumption of a single self-attention resource is reduced by 64.58%, and the number of model parameters and model size are reduced by more than 15%. By appropriately compressing the dimensions of the self-attention relationship variables, the Transformer network can be more efficient and even perform better. The results prompt us to rethink the reason why the self-attention mechanism works.

## 1 Introduction

The self-attention mechanism proposed by Vaswani et al. (2017) is originated from the natural language processing (NLP) Transformer network. Due to its superior performance, the self-attention mechanism has been widely used in the NLP field. The standard self-attention formula is defined as:

$$Attention(Q, K, V) = softmax(\frac{QK^T}{\sqrt{dim}})V \qquad (1)$$

The standard self-attention mechanism converts the input features linearly to obtain the variables Query ($Q$), Key ($K$), and Value ($V$) with the same dimension($dim$), then normalizes the result from the vector dot product of the $Q$ and $K$, and then obtains the potential relationship between every two features, finally uses this potential relationship to re-combine the features to get the output features. Compared to recurrent neural network (RNN) (Lipton et al., 2015) series, self-attention is characterized by directly calculating dependencies regardless of the distance between features and can be calculated in parallel. (Tang et al., 2018) demonstrate that the self-Attention mechanism is better at Word Sense Disambiguation (WSD) tasks(Raganato et al., 2017). The authors believe that the powerful semantic feature extraction ability is the key reason why self-Attention has better performance than RNN.

Most recent studies have focused on improving the efficiency of the self-attention mechanism as it brings the computational complexity of $O(N^2)$ to reach a considerable performance gain. Therefore, a batch of variants that pay more attention to computational efficiency are proposed on the

basis of original transformer backbone, such as LongFormer(Beltagy et al., 2020), Reformer(Kitaev et al., 2020), Faster-Transformer, Turbo-Transformers(Fang et al., 2020) etc.. Through changing the feature sequence, partial self-attention, hardware acceleration, and other methods, the Transformer model can be better used in the NLP field.

In the meanwhile, the excellent performance of the self-attention mechanism in the field of NLP has attracted great attention in computer vision. The Vision Transformer (ViT) backbone model proposed by(Dosovitskiy et al., 2020) divides the image information into patches for parallel input into the model, discards the decoder in the original transformer model, and retains its encoder as the feature extractor connected to the MLP layer to get output features. The results show that ViT is potentially competitive to the CNN network series(He et al., 2016), which increasingly makes the Self-Attention mechanism popular for visual tasks.

In addition to processing input and output features, ViT obeys the conventional Transformer Block, which makes ViT have high computational complexity, and coupled with the diversity of image feature information, it makes ViT training difficult. In essence, Transformer is data-hungry(Khan et al., 2021), which makes it a consensus to establish more efficient and extensive attention fusion image features. (Liu et al., 2021) proposes Swin Transformer, which further divides the features into several windows based on patch division. Swin introduces Non-Local neural network(Wang et al., 2018) to model the relationship of each patch in windows and realizes the cross-window relationship modeling through the shift window operation. Recently, Cswin(Dong et al., 2021) abandons the shift windows method, set windows as stripe, and models the horizontal and vertical relationships of features at the same time.

Even there are many similar applications of Transformer to the field of vision, but the works aim at optimizing the visual self-attention mechanism is relatively rare. This paper proposes a novel self-attention mechanism Low-relation Mutil-head Self-Attention(LMSA) to reduce the number of feature mapping $Query$ and $Key$ dimensions while maintaining the $Value$ dimension. This exploration breaks the information consistency of the traditional self-attention mechanism and greatly saves the computational complexity of a single self-Attention block. The diagram of LMSA is shown in Figure 1, which also illustrates the difference between the proposed LMSA and the newly proposed multi-head self-attention mechanism.

Specifically, we modified the self-attention mechanism based on recently proposed models such as Swin and Cswin, and we conducted a controlled experiment on the two attention mechanisms. In the Swin model, by appropriately compressing the number of dimensions, the image classification performance of the model has been further improved. In the Cswin model, we maximize the dimensionality compression of $Query$ and $Key$, while its performance on different datasets remains competitive to the traditional self-attention mechanism.

More importantly, through experiments, we demonstrate that the self-attention mechanism doesn't have to align the dimensions of $Query$, $Key$, and $Value$ to be same. The excessively high dimensions of $Query$ and $Key$ not only cause model data redundancy but even have a negative impact on the final model performance.

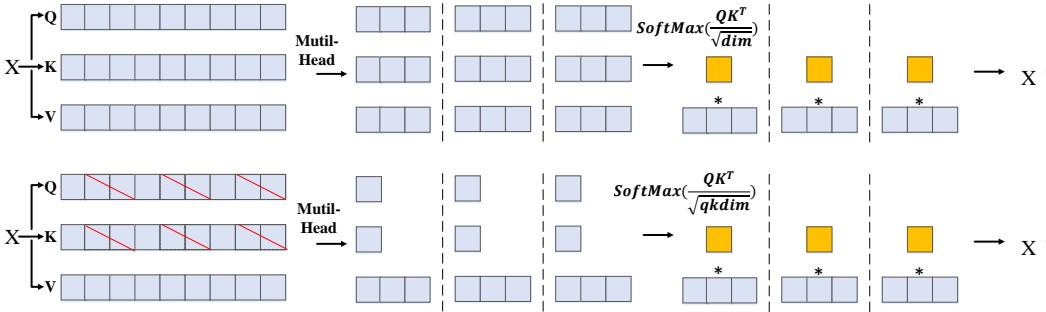

Figure 1: Comparison of LMSA and MSA operation process. Up: Standard multi-head self-attention mechanism, Down: Low-relation multi-head self-attention mechanism. qkdim represents the dimension of the Q or K variable after being compressed.

## 2 RELATED WORK

### 2.1 MUTIL-HEAD SELF-ATTENTION MACHINISM

For a long time, CNN and RNN as the first choice for sequence task models have the disadvantage that they cannot be calculated in parallel. The input of each subsequence depends on the output of the previous subsequence, and as the sequence length increases, it will be accompanied by long-term memory failure. The Transformer was proposed by (Vaswani et al., 2017). Transformer abandons the recurrent network structure and uses scaled dot-Product attention (see Equation 1) as its computing core to obtain the correlation of features at any two positions, and perform feature fusion based on this. In order to enable the model to pay more attention to the information of different representation subspaces at different positions, the author embeds the multi-head mechanism into self-attention. The formula of the multi-head self-attention mechanism is as follows:

$$MutilHead(Q, K, V) = Concat(head_1, ..., head_h)W^0$$
$$Where \quad head_i = Attention(QW_i^Q, KW_i^K, VW_i^V)$$

(2)

Where $W_i^Q \in \mathbb{R}^{d_{model} \times d_k}$ ,$W_i^K \in \mathbb{R}^{d_{model} \times d_k}$,$W_i^V \in \mathbb{R}^{d_{model} \times d_v}$,$W^O \in \mathbb{R}^{hd_v \times d_{model}}$.

Each head uses a different weight transformation, all heads are calculated in parallel, and the calculation results are concatenated to obtain the final output. Adjust the output scale by controlling the dimension of $Value$. Generally, the dimension of $Value$ is set to be the same as the input feature. After a complete multi-head attention calculation, the input and output sizes can be guaranteed to be consistent. It is convenient to embed various networks for multi-head attention.

### 2.2 EFFICIENT ATTENTION

The multi-head self-attention mechanism has achieved great success in the field of NLP , but it has also brought excessive complexity, especially in long text sequence tasks. Memory Compressed Transformer(Liu et al., 2018) tries to divide a long text sequence into several modules of similar size, and perform self-attention within the modules, and reduce the size of the self-attention matrix through stride convolution, so as to achieve the goal of improving efficiency. (Child et al., 2019) proposed the Sparse Transformer, which simplifies the dense attention to sparse attention through explicit selection, and improves the concentration of the model. Longformer (Beltagy et al., 2020) which based on Sparse Transformer combines sliding window and dilated sliding window to achieve local and global Balance of attention. Axial Transformer (Ho et al., 2019) applies multiple attention along a single axis of the input tensor. Since the length of any single axis is usually much smaller than the total number of elements, this model can significantly save computation and memory. Unlike the previously mentioned Efficient Transformer, Refomer(Kitaev et al., 2020) has a more in-depth discussion of the self-attention mechanism itself. By designing a reversible Transformer Block, Reformer greatly saves the space used to store intermediate results and proposed that locality sensitive hashing only calculates similar feature relationships, which effectively reduces the computational complexity. Synthesizer (Tay et al.) changes the self-attention mechanism more boldly. It directly abandoned the generation and calculation of $Query$ and $Key$, and instead used random initialization or linear projection of input features to directly replace the calculation results of $QK^T$. The author concludes through experiments that this method has advantages and disadvantages with the traditional self-attention mechanism, and has no affiliation with it, which makes people interested in the effective principle of the self-attention mechanism.

### 2.3 VISUAL TRANSFORMER

With the continuous development of Transformer in the NLP field, people have also begun to try to introduce the self-attention mechanism into visual tasks. After ViT(Dosovitskiy et al., 2020) was proposed, more efficient visual Transformers have also been proposed. Among them, the most concerned are Swin Tranformer(Liu et al., 2021) and Cswin Transformer(Dong et al., 2021). Swin Transformer realizes a more flexible local fusion of information through multi-level division of image features and then realizes information interaction between windows through shift operations. Cswin sets the windows in Swin as a bidirectional stripe with cross-feature capability, which is a successful application of Axial Attention in the computer vision field. Both Swin and CSwin adopted

the strategy proposed by PiT (Heo et al., 2021), dividing the entire model into several Layers, and down-sampling between the two layers. As the features move forward in the model, their scale decreases while their dimensions increase until the final High-Level feature is obtained to continue downstream tasks. The network structure of the Swin and Cswin model is shown in Figure 2.

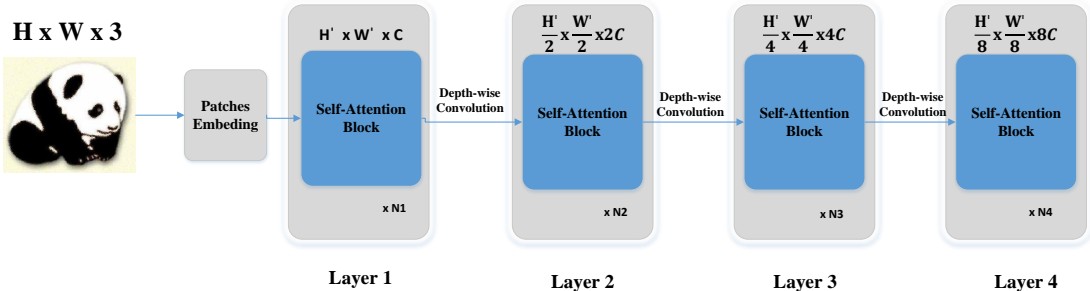

Figure 2: Overall structure of Swin and Cswin Transformer.

## 2.4 CONVOLUTIONAL VISION TRANSFORMER

Visual Tranformer's extensive research makes the Self-Attention mechanism often used to compare with convolution, Convolutional Vision Transformer(Wu et al., 2021) groundbreakingly combines convolution and self-attention mechanism. CvT discards position embeding and uses convolutional projection instead of linear projection to generate the Q, K, and V required for self-attention calculations, as shown in Figure 3. These changes enable CvT to well inherit the advantages of CNN while maintaining the dynamic self-encoding characteristics of Transformer: translation, scaling, and distortion invariance.

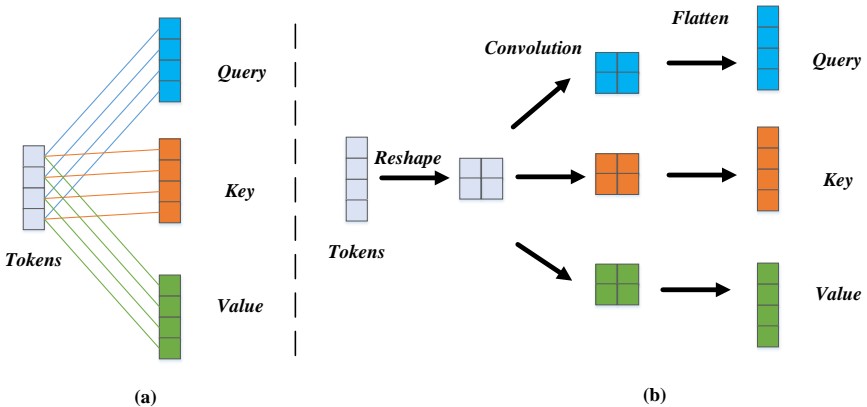

Figure 3: Comparison of Q, K, V generated by different self attention mechanisms.(a): Q, K, V generated by standard Self-Attention.(b): Q, K, V generated by CvT Self-Attention

## 3 METHODOLOGY

Unlike text sequence features, visual features often contain more nonobjective information. The development of Transformer in the computer vision field shows that the self-attention mechanism can effectively extract image features, and there is still much room for improvement. However, the standard self-attention mechanism has a computational complexity of $O(N^2)$. As the image size

increases, the self-attention mechanism consumes a lot more computing time and storage space, which limits the efficiency of the self-attention mechanism in visual tasks. To reduce the complexity of the self-attention mechanism and improve the computational efficiency, in this paper, we propose Low-Relation Mutil-Head Self-Attention, which aims to reduce the low-quality data that has a small effect on the calculation result in the self-attention.

## 3.1 LOW-RELATION MUTIL-HEAD SELF-ATTENTION

In order to solve the problem of the high complexity of the self-attention mechanism, a large number of Efficient Attentions have been proposed (Tay et al., 2020). The most common way to increase calculation speed is to control the number of $Keys$ and $Values$. Associate the $Query$ generated by the feature map with the $Key$ whose number is much smaller than $Query$, and act on the scaled $Value$ to achieve the purpose of improving efficiency by losing a certain accuracy. In the visual task, the most concerned optimization method is to adjust the receptive field of the self-attention mechanism to achieve a balance of accuracy and efficiency.

The above self-attention optimization methods mostly focus on adjusting the number of features used for calculation. From the perspective of feature dimensions, we try to compress the number of $Query$ and $Key$ dimensions as much as possible while keeping the $Value$ dimension unchanged and explore the impact of this compression on the accuracy of self-attention. In the proposed LMSA, the consistency of $Query$, $Key$, and $Value$ dimensions is destroyed. We control the overall complexity by adjusting the dimensions of $Query$ and $Key$. The core formula of LMSA is expressed as follows:

$$LMSA(Q, K, V) = MutilHead(CP(Q), CP(K), V) \tag{3}$$

where $CP(X)$ indicates the compress operation of $X$. In LMSA, we still use the multi-head mechanism to divide the feature into several heads, and each head is calculated in parallel. In the case of extreme compression, after $Query$ and $Key$ are divided into several heads, each head can only retain one dimension, that is, $dim_i^Q=1$. In specific tasks, we need to adjust the dimensions of $Query$ and $Key$ to meet the actual needs.

## 3.2 COMPLEXITY ANALYSIS

Minimizing the cost of $Query$ and $Key$ generation and the space occupied by them is the LMSA's core to improve the overall computing power. We take $dim^V$=64, $Num_{heads}$=2 as the calculation standard, and compare the optimization effects of self-attention complexity under different mapping methods, shown in Table 1. We use linear mapping or convolutional mapping to generate $Query$ and $Key$, the dimensions of $Query$ and $Key$ are the same as heads number in the case of extreme compression, $dim^Q$=2 and $dim^K$=2, which is much smaller than the standard self-attention mechanism $dim^Q$=64 and $dim^K$=64. Fewer dimensions make the calculation of the relationship matrix faster, and the complexity of the dot product between any two positions is reduced by 32 times, which can significantly reduce the complexity of the self-attention mechanism. In a single self-attention calculation process, when the extreme compression mode is used, time and space consumption are reduced by more than 60 percent.

$$\Omega(LMSA_{LINEAR}) = \Omega(Linear) + O(N^2 \cdot dim_{q,k} + N \cdot dim_v)$$
$$\Omega(LMSA_{CONV}) = \Omega(Conv) + O(N^2 \cdot dim_{q,k} + N \cdot dim_v) \tag{4}$$

## 3.3 DATA EFFICIENCY ANALYSIS

In the standard self-attention mechanism, the dimensions of $Query$, $Key$, and $Value$ are usually consistent, but we believe that this consistency is unnecessary. In self-attention calculations, the relationship between any two location features is derived from $Query$ and $Key$. The peak of the single-dimensional signal value of $Query$ and $Key$ has a crucial impact on the final result. The number of peaks that can be extracted from an element is limited. As the dimension increases, the $Query$ and $Key$ can not give more effective peak signals but will generate a large number of

Table 1: Complexity Analysis Between MSA And LMSA

| ATTENTION | PROJECTION | Q or K DIM | TOTAL OPERATION | SAVE(%) | PARAMS | SAVE(%) |
|---|---|---|---|---|---|---|
| stantard MSA | | 64 | 25354 | 0 | 12288 | 0 |
| LMSA | Linear | 32 | 16906 | 33.32 | 8192 | 33.33 |
| | | 16 | 12682 | 49.98 | 6144 | 50 |
| LMSA(extreme mode) | | 2 | **8986** | **64.56** | **4352** | **64.58** |
| stantard MSA | | 64 | 14.38M | 0 | 110592 | 0 |
| LMSA | Convolution (kernel size:3x3) | 32 | 9.58M | 33.38 | 73728 | 33.33 |
| | | 16 | 7.19M | 50 | 55296 | 50 |
| LMSA(extreme mode) | | 2 | **5.09M** | **64.6** | **39168** | **64.58** |

low-quality signals. A single low-quality signal effect on the results is very limited, but the high proportion of low-quality signals will cause a lot of wasted space, and make the training slow, even interfere with peak signals. Instead, we also found in experiments that too few dimensions of $Query$ and $Key$ will also cause the loss of key peaks, which will have a destructive effect. Properly adjusting the dimensions of $Query$ and $Key$ can reduce the overall complexity while maintaining the original accuracy or even improving accuracy. To explore the data efficiency of $Query$ and $Key$ at a deeper level, we take the Swin model as an example, use the standard self-attention mechanism and LMSA for training respectively, and count the data efficiency of Query and Key at different features from low-level to high-level. We use four thresholds of 0.1, 0.5, 1, and 2 to calculate the proportion of signals whose absolute value exceeds these thresholds in $Query$ and $Key$ to the total number of signals. As shown in Figure 4, LMSA is more efficient than MSA.

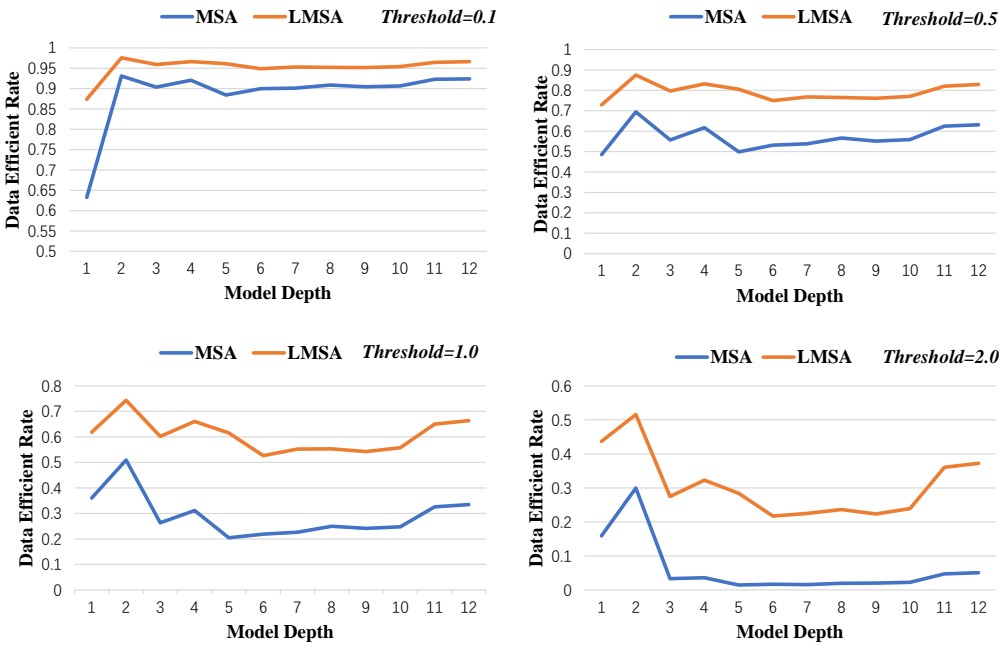

Figure 4: Data effcient analysis of MSA and LMSA at different thresholds

Table 2: Comparison of complexity between different efficient attention

| EFFCIENT ATTENTION | Complexity |
|---|---|
| Memory Compressed Attention | $O(N^2 \cdot D)$ |
| Sparse Attention | $O(N\sqrt{N} \cdot D)$ |
| Longformer | $O(N \cdot (k + g) \cdot D)$ |
| Axial Attention | $O(N \cdot (H + W) \cdot D)$ |
| Refomer | $O(N \cdot log(N) \cdot D)$ |
| Synthesizer | $O(N^2 \cdot D)$ |
| LMSA | $\mathbf{O(N^2 \cdot D_{q,k})}$ |

### 3.4 COMPARISON WITH OTHER EFFICIENT ATTENTION

When using Transformer for image feature extraction tasks, in order to reintegrate feature information and reduce feature scales, it is often used in conjunction with downsampling, which makes High-Level features have more dimensions, this also shows that it is necessary to optimize the number of dimensions for visual tasks. Table 2 lists the comparison of the computational complexity of the main Efficient Transformer, the work of dimensional optimization is very rare among them. From the perspective of the number of dimensions, LMSA can reduce the computational complexity more conveniently and effectively.

## 4 EXPERIMENTS

### 4.1 IMAGE CLASSIFICATION

In order to verify the effectiveness of the method, we conduct comparative experiments on the currently common Visual Transformer to get the results. Taking into account the different mapping methods of the self-attention mechanism, we selected linear correlation self-attention models: Swin-Transformer and Cswin Transformer, and also selected convolutional self-attention models: CvT, And perform image classification training on CIFAR(Krizhevsky et al., 2009) and Caltech(Griffin et al., 2007) datasets. Without quoting any external data for pre-training, we got the results and analyzed them. The experimental results are shown in Table 3.

Both Swin(Liu et al., 2021)and Cswin(Dong et al., 2021) adopt a similar overall architecture based on PiT(Heo et al., 2021). Divide the picture into several patches for embedding firstly and then do subsequent calculations. The overall model is divided into four layers. Down-sampling operations are assembled between layers. The layers are composed of multiple self-Attention Blocks and MLPs. The overall architecture is shown in Figure 2. We replaced all internal MSAs with LMSA and got the modified model.

We train on the CIFAR dataset firstly. CIFAR-10 and CIFAR-100 are subsets of the 80 million labeled tiny image datasets, and their image size is 32x32. Because the image size is small enough, the Patches Embedding stage of the model is replaced by Conv 1x1, which can preserve the image information to the maximum extent and adapt the model. We have followed the Cswin training strategy, using the MSA and LMSA methods to train 300 epochs on both the Swin and CSwin models. We use Adamw(Loshchilov & Hutter, 2017) as the optimizer, and the weight decay is set to 0.05. The warmup(Goyal et al., 2017) method is used to initialize the learning rate and the learning rate is set to 0.0005, and the Cosine Annealing(Loshchilov & Hutter, 2016) method is used to adjust the learning rate. In order to prevent overfitting, we added Dropout with a probability of 0.1 inside each Transformer Block. In data processing, we combined MixUp(Zhang et al., 2017) and CutMix(Yun et al., 2019) data augmentation methods, and set label smoothing(Szegedy et al., 2016) to 0.1. Finally, we use soft target cross-entropy to calculate the loss between the label and the prediction result.

In order to further verify that LMSA can adapt well to datasets with different digital features, we also conducted experiments on the Caltech dataset. The Caltech101 and Caltech-256 datasets have the characteristics of uneven distribution of categories, and their image sizes are not fixed, which can better test the performance of the model. In data processing, we fixed the pictures to 224x224 by

Table 3: Comparison of image classification results between MSA and LMSA

| Model | Dataset | Method | Q or K Dim | Params(M) | Model Size(M) | Acc(%) |
|-------|---------|--------|------------|-----------|---------------|--------|
| CSwin | CIFAR10 | MSA | [64,128,256,512] | 21.8 | 262.1 | **95.25** |
| | | LMSA(extreme) | [2,4,8,16] | **18.65** | **224.4** | 95.24 |
| | CIFAR100 | MSA | [64,128,256,512] | 21.85 | 262.6 | **80.28** |
| | | LMSA(extreme) | [2,4,8,16] | **18.69** | **224.5** | 80.09 |
| | Caltech101 | MSA | [64,128,256,512] | 21.85 | 262.7 | 73.62 |
| | | LMSA(extreme) | [2,4,8,16] | **18.7** | **225** | **73.63** |
| | Caltech256 | MSA | [64,128,256,512] | 21.86 | 263.7 | 70.3 |
| | | LMSA(extreme) | [2,4,8,16] | **18.7** | **226** | **71.71** |
| Swin | CIFAR10 | MSA | [96,192,384,768] | 27.52 | 331.1 | 93.21 |
| | | LMSA(extreme) | [3,6,12,24] | **23.33** | **280.9** | **93.64** |
| | CIFAR100 | MSA | [96,192,384,768] | 27.59 | 331.9 | 77 |
| | | LMSA | [48,96,192,384] | 25.43 | 306.1 | **77.38** |
| | | LMSA(extreme) | [3,6,12,24] | **23.40** | **281.7** | 75.46 |
| CvT | CIFAR10 | MSA | [64,192,384] | 61.4 | 234.5 | **85.36** |
| | | LMSA(extreme) | [1,3,6] | **30.8** | **117.6** | 85.24 |
| | CIFAR100 | MSA | [64,192,384] | 61.4 | 246.1 | **63.83** |
| | | LMSA(extreme) | [1,3,6] | **30.8** | **123.5** | 63.48 |

random cropping method, set the patch size to 4 for Patches Embedding, obtained 56x56 features, and input them into the network for training.

CvT changes the linear correlation in self-attention to 3x3 convolution, which makes it necessary to test LSMA on CvT. We initialize the learning rate to 0.02 and train 300 epoch with a cosine learning rate decay scheduler.

On the CIFAR-10 dataset, using the Q, K extreme compression strategy, the performance of CSwin and CvT with LSMA is basically the same as with MSA, and the number of parameters has been greatly reduced. Especially in CvT, both the parameter number and the final model size are reduced by more than 50 percent. In the Swin model, using extreme compression LMSA even achieved higher accuracy than MSA, which shows that controlling the proportion of low-quality data is conducive to model convergence. On the CIFAR-100 dataset, LMSA extreme compression brings a slight accuracy loss. We also appropriately increase dimensions, for example, on Swin, we only perform dimension compression by 2 times, which also exceeds MSA.

We are also training on the Caltech dataset, especially on the Caltech-256 dataset,self-attention with extreme compression shows stronger performance than MSA, which further shows that self-attention data redundancy is harmful to the model.

## 4.2 SEMANTIC SEGMENTATION

In addition to image classification tasks, Transformer can also be well applied to other tasks, such as semantic segmentation. Segformer(Xie et al., 2021) is a simple, efficient, and powerful semantic segmentation framework. It combines Transformer with MLP, uses the Self-Attention mechanism as a feature extractor to improve the effective receptive field(Luo et al., 2016), and uses MLP as a feature decoder. Segfomer also uses Efficient attention to control the complexity of the overall self-attention mechanism by establishing a one-to-many relationship between $Query$ and $Key$. We test the performance of LSMA on ADE20K(Zhou et al., 2017) without pre-training by introducing it to Segformer. The results are shown in Table 4. The Segformer using LMSA has the same effect as the original Segformer, which also shows that LSMA can be used in conjunction with other Efficient Attention to achieve better results.

## 4.3 ABILITION STUDY

We design ablation experiments based on the Swin model, starting with Standard self-attention, gradually compressing the Q and K dimensions of self-attention, using the same hyperparameter

Table 4: Comparison of semantic segmentation results between MSA and LMSA

| Model | Method | Q or K Dim | mIoU(%) | mACC(%) |
|---|---|---|---|---|
| SegFormer | MSA | [32,64,160,256] | **22.97** | 31.74 |
| | LMSA | [24,48,120,192] | 22.95 | **31.82** |
| | | [16,32,80,128] | 22.87 | 31.79 |

settings for CIFAR-100 image classification training, and observing the results. We compressed 2, 4, 8, 16, and 32 times (extreme) respectively. We found that the compression of the model is accompanied by an increase in data efficiency. After 2 times and 4 times compression, the final performance of the model has been improved. But after 16,32 times compression, the accuracy of the model is destroyed. This shows that it is not that the lower the dimensionality, the more beneficial the model. Too low $Query$ and $Key$ dimensionality will cause high-quality signal loss and affect the overall performance of the model.

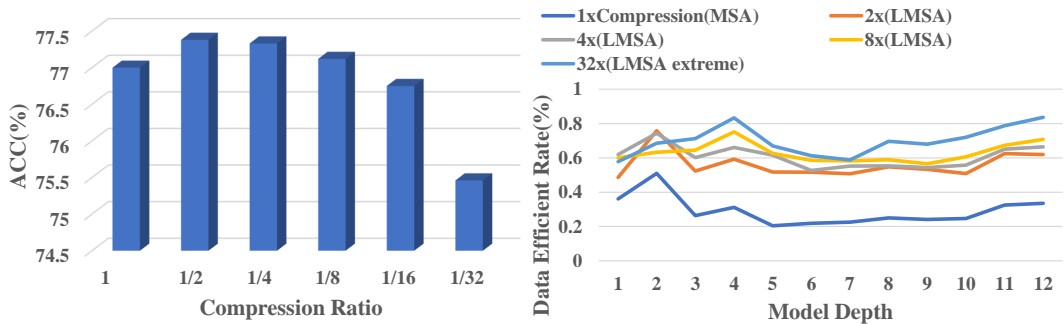

Figure 5: Effect of different compression ratio on results.Cifar-100 is used to experiment on Swin transformer without pre-training.Left:Influence of different compression ratio on accuracy.Right:Data effective rate under different compression rates

## 5 CONCLUSION

This paper proposes a novel variant of Efficient Attention: LMSA, which limits the number of signals that generate relational matrices, improves the quality of $Query$ and $Key$ signals and greatly reduces the computational consumption of MSA. Experiments have verified that LMSA can be equal to MSA in accuracy and even has advantages. This brings the confidence to LMSA instead of MSA to embed various types of Transformers. We demonstrate that dimensional consistency is unnecessary in self-attention calculations, choosing an appropriate LMSA can achieve competitive results while saving computing resources.

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
