# OpenReview forum: "LMSA: Low-relation Mutil-head Self-Attention Mechanism in Visual Transformer"
_ICLR.cc/2022/Conference — ICLR 2022 Submitted_

### Official Review · Reviewer_S5RZ · 2021-10-30

**Correctness:** 2
**Technical Novelty And Significance:** 1
**Empirical Novelty And Significance:** 1
**Recommendation:** 3
**Confidence:** 5

**Main Review:**

Strengths:
The paper is simple, easy to understand, and applicable to most self-attention operators.

Weaknesses:
1. Technical novelty is very limited. Using fewer channels somewhere in a neural network can hardly be a contribution to the community. Furthermore, [1] searches for the best number of channels for query and key.
2. Experiments are done mostly on very small datasets and the performance is usually worse with the proposed LMSA. The conclusion may not be scalable.
3. Data efficiency usually refers to training with fewer data points/samples, but in section 3.3, the authors measure something like activation norm/peak/temperature (no equation is given) and refer to it as data efficiency. However, it is expected that two models with different query/key channels can have different norm/peak/temperature, even at random initialization.

[1] Neural Architecture Search for Lightweight Non-Local Networks. CVPR 2020.

**Summary Of The Paper:**

The authors propose to use fewer channels for query and key in multi-head self-attention and claim that the choice is better for CIFAR, Caltech, etc.

**Summary Of The Review:**

I think this paper is not good enough mainly due to the very limited novelty and results.

---

### Official Review · Reviewer_ZniF · 2021-10-30

**Correctness:** 3
**Technical Novelty And Significance:** 1
**Empirical Novelty And Significance:** 1
**Recommendation:** 1
**Confidence:** 5

**Details Of Ethics Concerns:**

No.

**Main Review:**

Strengths:
1. This paper is well-motivated since self-attention computation is the critical issue for Transformer.
2. The experimental results demonstrate the effectiveness of the proposed method.

Weaknesses:
1. novelty concerns:

(a) The main issue is the novelty. This paper mainly proposes to reduce the dimensions of Query and Key. However, this looks like a hyperparameter tuning for the dimension numbers unless the paper proposes a novel way to compress the dimension, which is missing in this paper.

(b) Moreover, the spatial-wise computation is the main bottleneck for self-attention. This paper does not address this issue. Table 2 shows that the complexity is still O(N^2). It would be much better to explain this or provide proof showing that spatial-wise computation is not an issue for the problem that the paper is addressing.


2. claim concerns:

(a) The paper claims that “the works aim at optimizing the visual self-attention mechanism is relatively rare”, but it is not true. For example, there are papers working on changing computing orders of Queries, Keys, and Values, inside self-attention, like ET [1], LambdaNetworks [2]. Some papers work on reducing spatial computation of self-attention, such as HaloNet [4]. Moreover, LRA [3] proposed a new benchmark including vision tasks to investigate several state-of-the-art self-attention mechanisms.

References:

[1] Zhuoran Shen, et al. "Efficient Attention: Attention with Linear Complexities", WACV, 2021.

[2] Irwan Bello. "LambdaNetworks: Modeling long-range Interactions without Attention", ICLR, 2021.

[3] Yi Tay, et al. "Long Range Arena: A Benchmark for Efficient Transformers", ICLR, 2021.

[4] Ashish Vaswani, et al. "Scaling Local Self-Attention for Parameter Efficient Visual Backbones", CVPR, 2021.

(b) The paper claims that “we demonstrate that the self-attention mechanism doesn’t have to align the dimensions of Query, Key, and Value to be same”. However, this result is not surprising. Adjusting dimension numbers for Query, Key, and Value is the way to find the trade-off between computation and accuracy.


3. technical detail concerns:

(a) It is not clear why the paper chooses to reduce the dimensions of Query and Key instead of Key and Value like most previous work. It would be great to provide more analyses to support this decision.

(b) The paper only mentions CP(X) is the compress operation of X, but does not describe technical details of CP(X).


4. experiment concerns:

(a) The experimental results shown in Tables 3 and 4 do not show significant improvement of LMSA over MSA. Moreover, there are no experiments on ImageNet like most previous work does, which makes the experimental results not convincing.


**Summary Of The Paper:**

This paper proposes an efficient self-attention mechanism which compresses the channel numbers of features, to reduce the computation for the Transformer.

**Summary Of The Review:**

The proposed method lacks novelty, and the technical details of the proposed method are missing. Moreover, the proposed method is not evaluated on a common benchmark dataset (i.e., ImageNet) and the results are not impressive. Therefore, I recommend rejecting this paper.

---

### Official Review · Reviewer_kY4w · 2021-11-06

**Correctness:** 2
**Technical Novelty And Significance:** 1
**Empirical Novelty And Significance:** 1
**Recommendation:** 3
**Confidence:** 5

**Main Review:**

Pros:
The proposed idea is simple and technically soundable. The writing is good and the organization is easy to follow. They also integrated  LMSA with various existing transformer-based networks and evaluated on several benchmarks.

Cons:
1. Novelty & Significance are limited. The proposed idea is too natural and straightforward, while similar techniques (or tricks) has already been used in many existing approaches/practices. So I did not think the novelty/significance of this work meets the accepted criteria
2. The experimental validation was only conducted on the weak baseline or small datasets. For image classification, the validation on ImageNet is missed, and the reported performance on semantic segmentation is too weak. These weaknesses make it is hard to justify the effectiveness of LMSA.
3. The self-attention itself may not be the bottleneck of many existing transformer-based self-attention, such as Swin & CSwin. In fact, the FFN is the bottleneck. Therefore, I really wonder if the proposed method could accelerate the inference speed, and it is not reported in the paper.

**Summary Of The Paper:**

This paper presents a mechanism to reduce the computation costs of a standard self-attention module, named LMSA.
The basic idea of LMSA is to reduce the dimension of key&query of self-attention(SA) while keeping the dimension of value unchanged. Therefore, the computational complexity of SA will be reduced from O(N^2D) to O(N^2D_q,k).
They study the proposed mechanism on image classification and semantic segmentation.


**Summary Of The Review:**

This work lacks novelty and significance. The experiments also have many drawbacks, such as only validating on the weak baseline or small dataset.

---

### Note · Authors · 2024-04-12
**Submission Withdrawn by the Authors**

I have read and agree with the venue's withdrawal policy on behalf of myself and my co-authors.

---

### Decision · Program_Chairs · 2022-01-20

**Decision:**

Reject

**Comment:**

This paper proposes a simple change to Transformer architecture to improve efficiency. While the reviewers appreciate the writing, all the reviewers agree that the novelty and contributions of the paper are limited both in the problem being solved by the paper and the level of experiments in it. Authors did not respond to reviewer's comments. Hence I recommend rejection.